# Convolutional LSTM Network: A Machine Learning Approach for Precipitation Nowcasting

**Xingjian Shi    Zhourong Chen    Hao Wang    Dit-Yan Yeung**
Department of Computer Science and Engineering
Hong Kong University of Science and Technology
{xshiab,zchenbb,hwangaz,dyyeung}@cse.ust.hk

**Wai-kin Wong    Wang-chun Woo**
Hong Kong Observatory
Hong Kong, China
{wkwong,wcwoo}@hko.gov.hk

## Abstract

The goal of precipitation nowcasting is to predict the future rainfall intensity in a local region over a relatively short period of time. Very few previous studies have examined this crucial and challenging weather forecasting problem from the machine learning perspective. In this paper, we formulate precipitation nowcasting as a spatiotemporal sequence forecasting problem in which both the input and the prediction target are spatiotemporal sequences. By extending the *fully connected LSTM* (FC-LSTM) to have convolutional structures in both the input-to-state and state-to-state transitions, we propose the *convolutional LSTM* (ConvLSTM) and use it to build an end-to-end trainable model for the precipitation nowcasting problem. Experiments show that our ConvLSTM network captures spatiotemporal correlations better and consistently outperforms FC-LSTM and the state-of-the-art operational ROVER algorithm for precipitation nowcasting.

## 1   Introduction

Nowcasting convective precipitation has long been an important problem in the field of weather forecasting. The goal of this task is to give precise and timely prediction of rainfall intensity in a local region over a relatively short period of time (e.g., 0-6 hours). It is essential for taking such timely actions as generating society-level emergency rainfall alerts, producing weather guidance for airports, and seamless integration with a longer-term numerical weather prediction (NWP) model. Since the forecasting resolution and time accuracy required are much higher than other traditional forecasting tasks like weekly average temperature prediction, the precipitation nowcasting problem is quite challenging and has emerged as a hot research topic in the meteorology community [22].

Existing methods for precipitation nowcasting can roughly be categorized into two classes [22], namely, NWP based methods and radar echo[1] extrapolation based methods. For the NWP approach, making predictions at the nowcasting timescale requires a complex and meticulous simulation of the physical equations in the atmosphere model. Thus the current state-of-the-art operational precipitation nowcasting systems [19, 6] often adopt the faster and more accurate extrapolation based methods. Specifically, some computer vision techniques, especially optical flow based methods, have proven useful for making accurate extrapolation of radar maps [10, 6, 20]. One recent progress along this path is the *Real-time Optical flow by Variational methods for Echoes of Radar* (ROVER)

algorithm [25] proposed by the Hong Kong Observatory (HKO) for its *Short-range Warning of Intense Rainstorms in Localized System* (SWIRLS) [15]. ROVER calculates the optical flow of consecutive radar maps using the algorithm in [5] and performs semi-Lagrangian advection [4] on the flow field, which is assumed to be still, to accomplish the prediction. However, the success of these optical flow based methods is limited because the flow estimation step and the radar echo extrapolation step are separated and it is challenging to determine the model parameters to give good prediction performance.

These technical issues may be addressed by viewing the problem from the machine learning perspective. In essence, precipitation nowcasting is a spatiotemporal sequence forecasting problem with the sequence of past radar maps as input and the sequence of a fixed number (usually larger than 1) of future radar maps as output.[2] However, such learning problems, regardless of their exact applications, are nontrivial in the first place due to the high dimensionality of the spatiotemporal sequences especially when multi-step predictions have to be made, unless the spatiotemporal structure of the data is captured well by the prediction model. Moreover, building an effective prediction model for the radar echo data is even more challenging due to the chaotic nature of the atmosphere.

Recent advances in deep learning, especially recurrent neural network (RNN) and long short-term memory (LSTM) models [12, 11, 7, 8, 23, 13, 18, 21, 26], provide some useful insights on how to tackle this problem. According to the philosophy underlying the deep learning approach, if we have a reasonable end-to-end model and sufficient data for training it, we are close to solving the problem. The precipitation nowcasting problem satisfies the data requirement because it is easy to collect a huge amount of radar echo data continuously. What is needed is a suitable model for end-to-end learning. The pioneering LSTM encoder-decoder framework proposed in [23] provides a general framework for sequence-to-sequence learning problems by training temporally concatenated LSTMs, one for the input sequence and another for the output sequence. In [18], it is shown that prediction of the next video frame and interpolation of intermediate frames can be done by building an RNN based language model on the visual words obtained by quantizing the image patches. They propose a recurrent convolutional neural network to model the spatial relationships but the model only predicts one frame ahead and the size of the convolutional kernel used for state-to-state transition is restricted to 1. Their work is followed up later in [21] which points out the importance of multi-step prediction in learning useful representations. They build an LSTM encoder-decoder-predictor model which reconstructs the input sequence and predicts the future sequence simultaneously. Although their method can also be used to solve our spatiotemporal sequence forecasting problem, the *fully connected LSTM* (FC-LSTM) layer adopted by their model does not take spatial correlation into consideration.

In this paper, we propose a novel *convolutional LSTM* (ConvLSTM) network for precipitation nowcasting. We formulate precipitation nowcasting as a spatiotemporal sequence forecasting problem that can be solved under the general sequence-to-sequence learning framework proposed in [23]. In order to model well the spatiotemporal relationships, we extend the idea of FC-LSTM to ConvLSTM which has convolutional structures in both the input-to-state and state-to-state transitions. By stacking multiple ConvLSTM layers and forming an encoding-forecasting structure, we can build an end-to-end trainable model for precipitation nowcasting. For evaluation, we have created a new real-life radar echo dataset which can facilitate further research especially on devising machine learning algorithms for the problem. When evaluated on a synthetic Moving-MNIST dataset [21] and the radar echo dataset, our ConvLSTM model consistently outperforms both the FC-LSTM and the state-of-the-art operational ROVER algorithm.

## 2 Preliminaries

### 2.1 Formulation of Precipitation Nowcasting Problem

The goal of precipitation nowcasting is to use the previously observed radar echo sequence to forecast a fixed length of the future radar maps in a local region (e.g., Hong Kong, New York, or Tokyo). In real applications, the radar maps are usually taken from the weather radar every 6-10 minutes and nowcasting is done for the following 1-6 hours, i.e., to predict the 6-60 frames ahead. From the ma-

chine learning perspective, this problem can be regarded as a spatiotemporal sequence forecasting problem.

Suppose we observe a dynamical system over a spatial region represented by an $M \times N$ grid which consists of $M$ rows and $N$ columns. Inside each cell in the grid, there are $P$ measurements which vary over time. Thus, the observation at any time can be represented by a tensor $\mathcal{X} \in \mathbf{R}^{P \times M \times N}$, where $\mathbf{R}$ denotes the domain of the observed features. If we record the observations periodically, we will get a sequence of tensors $\hat{\mathcal{X}}_1, \hat{\mathcal{X}}_2, \ldots, \hat{\mathcal{X}}_t$. The spatiotemporal sequence forecasting problem is to predict the most likely length-$K$ sequence in the future given the previous $J$ observations which include the current one:

$$\tilde{\mathcal{X}}_{t+1}, \ldots, \tilde{\mathcal{X}}_{t+K} = \underset{\mathcal{X}_{t+1}, \ldots, \mathcal{X}_{t+K}}{\arg \max} \; p(\mathcal{X}_{t+1}, \ldots, \mathcal{X}_{t+K} \mid \hat{\mathcal{X}}_{t-J+1}, \hat{\mathcal{X}}_{t-J+2}, \ldots, \hat{\mathcal{X}}_t) \qquad (1)$$

For precipitation nowcasting, the observation at every timestamp is a 2D radar echo map. If we divide the map into tiled non-overlapping patches and view the pixels inside a patch as its measurements (see Fig. 1), the nowcasting problem naturally becomes a spatiotemporal sequence forecasting problem.

We note that our spatiotemporal sequence forecasting problem is different from the one-step time series forecasting problem because the prediction target of our problem is a sequence which contains both spatial and temporal structures. Although the number of free variables in a length-$K$ sequence can be up to $O(M^K N^K P^K)$, in practice we may exploit the structure of the space of possible predictions to reduce the dimensionality and hence make the problem tractable.

## 2.2 Long Short-Term Memory for Sequence Modeling

For general-purpose sequence modeling, LSTM as a special RNN structure has proven stable and powerful for modeling long-range dependencies in various previous studies [12, 11, 17, 23]. The major innovation of LSTM is its memory cell $c_t$ which essentially acts as an accumulator of the state information. The cell is accessed, written and cleared by several self-parameterized controlling gates. Every time a new input comes, its information will be accumulated to the cell if the input gate $i_t$ is activated. Also, the past cell status $c_{t-1}$ could be "forgotten" in this process if the forget gate $f_t$ is on. Whether the latest cell output $c_t$ will be propagated to the final state $h_t$ is further controlled by the output gate $o_t$. One advantage of using the memory cell and gates to control information flow is that the gradient will be trapped in the cell (also known as constant error carousels [12]) and be prevented from vanishing too quickly, which is a critical problem for the vanilla RNN model [12, 17, 2]. FC-LSTM may be seen as a multivariate version of LSTM where the input, cell output and states are all 1D vectors. In this paper, we follow the formulation of FC-LSTM as in [11]. The key equations are shown in (2) below, where '∘' denotes the Hadamard product:

$$\begin{aligned}
i_t &= \sigma(W_{xi} x_t + W_{hi} h_{t-1} + W_{ci} \circ c_{t-1} + b_i) \\
f_t &= \sigma(W_{xf} x_t + W_{hf} h_{t-1} + W_{cf} \circ c_{t-1} + b_f) \\
c_t &= f_t \circ c_{t-1} + i_t \circ \tanh(W_{xc} x_t + W_{hc} h_{t-1} + b_c) \\
o_t &= \sigma(W_{xo} x_t + W_{ho} h_{t-1} + W_{co} \circ c_t + b_o) \\
h_t &= o_t \circ \tanh(c_t)
\end{aligned} \qquad (2)$$

Multiple LSTMs can be stacked and temporally concatenated to form more complex structures. Such models have been applied to solve many real-life sequence modeling problems [23, 26].

# 3 The Model

We now present our ConvLSTM network. Although the FC-LSTM layer has proven powerful for handling temporal correlation, it contains too much redundancy for spatial data. To address this problem, we propose an extension of FC-LSTM which has convolutional structures in both the input-to-state and state-to-state transitions. By stacking multiple ConvLSTM layers and forming an encoding-forecasting structure, we are able to build a network model not only for the precipitation nowcasting problem but also for more general spatiotemporal sequence forecasting problems.

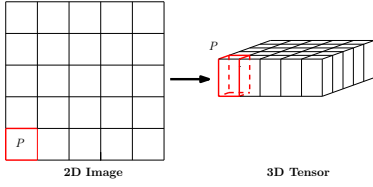

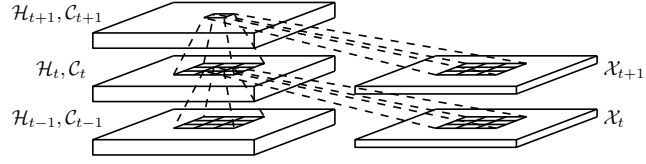

Figure 1: Transforming 2D image into 3D tensor

Figure 2: Inner structure of ConvLSTM

### 3.1 Convolutional LSTM

The major drawback of FC-LSTM in handling spatiotemporal data is its usage of full connections in input-to-state and state-to-state transitions in which no spatial information is encoded. To overcome this problem, a distinguishing feature of our design is that all the inputs $\mathcal{X}_1, \ldots, \mathcal{X}_t$, cell outputs $\mathcal{C}_1, \ldots, \mathcal{C}_t$, hidden states $\mathcal{H}_1, \ldots, \mathcal{H}_t$, and gates $i_t, f_t, o_t$ of the ConvLSTM are 3D tensors whose last two dimensions are spatial dimensions (rows and columns). To get a better picture of the inputs and states, we may imagine them as vectors standing on a spatial grid. The ConvLSTM determines the future state of a certain cell in the grid by the inputs and past states of its local neighbors. This can easily be achieved by using a convolution operator in the state-to-state and input-to-state transitions (see Fig. 2). The key equations of ConvLSTM are shown in (3) below, where '$*$' denotes the convolution operator and '$\circ$', as before, denotes the Hadamard product:

$$
\begin{aligned}
i_t &= \sigma(W_{xi} * \mathcal{X}_t + W_{hi} * \mathcal{H}_{t-1} + W_{ci} \circ \mathcal{C}_{t-1} + b_i) \\
f_t &= \sigma(W_{xf} * \mathcal{X}_t + W_{hf} * \mathcal{H}_{t-1} + W_{cf} \circ \mathcal{C}_{t-1} + b_f) \\
\mathcal{C}_t &= f_t \circ \mathcal{C}_{t-1} + i_t \circ \tanh(W_{xc} * \mathcal{X}_t + W_{hc} * \mathcal{H}_{t-1} + b_c) \\
o_t &= \sigma(W_{xo} * \mathcal{X}_t + W_{ho} * \mathcal{H}_{t-1} + W_{co} \circ \mathcal{C}_t + b_o) \\
\mathcal{H}_t &= o_t \circ \tanh(\mathcal{C}_t)
\end{aligned}
\tag{3}
$$

If we view the states as the hidden representations of moving objects, a ConvLSTM with a larger transitional kernel should be able to capture faster motions while one with a smaller kernel can capture slower motions. Also, if we adopt a similar view as [16], the inputs, cell outputs and hidden states of the traditional FC-LSTM represented by (2) may also be seen as 3D tensors with the last two dimensions being 1. In this sense, FC-LSTM is actually a special case of ConvLSTM with all features standing on a single cell.

To ensure that the states have the same number of rows and same number of columns as the inputs, padding is needed before applying the convolution operation. Here, padding of the hidden states on the boundary points can be viewed as using the *state of the outside world* for calculation. Usually, before the first input comes, we initialize all the states of the LSTM to zero which corresponds to "total ignorance" of the future. Similarly, if we perform zero-padding (which is used in this paper) on the hidden states, we are actually setting the *state of the outside world* to zero and assume no prior knowledge about the outside. By padding on the states, we can treat the boundary points differently, which is helpful in many cases. For example, imagine that the system we are observing is a moving ball surrounded by walls. Although we cannot see these walls, we can infer their existence by finding the ball bouncing over them again and again, which can hardly be done if the boundary points have the same state transition dynamics as the inner points.

### 3.2 Encoding-Forecasting Structure

Like FC-LSTM, ConvLSTM can also be adopted as a building block for more complex structures. For our spatiotemporal sequence forecasting problem, we use the structure shown in Fig. 3 which consists of two networks, an encoding network and a forecasting network. Like in [21], the initial states and cell outputs of the forecasting network are copied from the last state of the encoding network. Both networks are formed by stacking several ConvLSTM layers. As our prediction target has the same dimensionality as the input, we concatenate all the states in the forecasting network and feed them into a $1 \times 1$ convolutional layer to generate the final prediction.

We can interpret this structure using a similar viewpoint as [23]. The *encoding* LSTM compresses the whole input sequence into a hidden state tensor and the *forecasting* LSTM unfolds this hidden

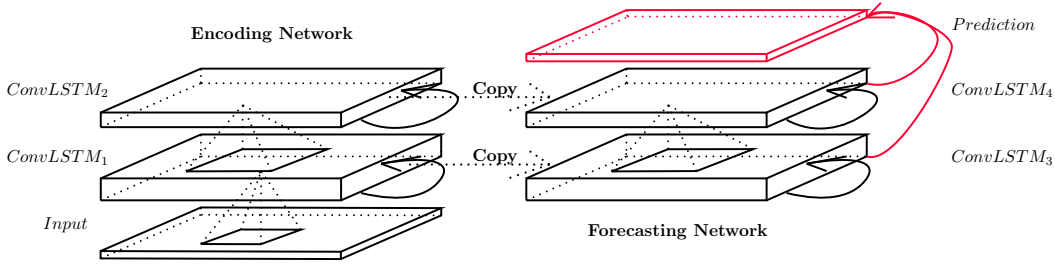

Figure 3: Encoding-forecasting ConvLSTM network for precipitation nowcasting

state to give the final prediction:

$$
\begin{aligned}
\tilde{\mathcal{X}}_{t+1}, \ldots, \tilde{\mathcal{X}}_{t+K} &= \underset{\mathcal{X}_{t+1}, \ldots, \mathcal{X}_{t+K}}{\arg \max} \; p(\mathcal{X}_{t+1}, \ldots, \mathcal{X}_{t+K} \mid \hat{\mathcal{X}}_{t-J+1}, \hat{\mathcal{X}}_{t-J+2}, \ldots, \hat{\mathcal{X}}_{t}) \\
&\approx \underset{\mathcal{X}_{t+1}, \ldots, \mathcal{X}_{t+K}}{\arg \max} \; p(\mathcal{X}_{t+1}, \ldots, \mathcal{X}_{t+K} \mid f_{encoding}(\hat{\mathcal{X}}_{t-J+1}, \hat{\mathcal{X}}_{t-J+2}, \ldots, \hat{\mathcal{X}}_{t})) \quad (4) \\
&\approx g_{forecasting}(f_{encoding}(\hat{\mathcal{X}}_{t-J+1}, \hat{\mathcal{X}}_{t-J+2}, \ldots, \hat{\mathcal{X}}_{t}))
\end{aligned}
$$

This structure is also similar to the LSTM future predictor model in [21] except that our input and output elements are all 3D tensors which preserve all the spatial information. Since the network has multiple stacked ConvLSTM layers, it has strong representational power which makes it suitable for giving predictions in complex dynamical systems like the precipitation nowcasting problem we study here.

## 4  Experiments

We first compare our ConvLSTM network with the FC-LSTM network on a synthetic Moving-MNIST dataset to gain some basic understanding of the behavior of our model. We run our model with different number of layers and kernel sizes and also study some "out-of-domain" cases as in [21]. To verify the effectiveness of our model on the more challenging precipitation nowcasting problem, we build a new radar echo dataset and compare our model with the state-of-the-art ROVER algorithm based on several commonly used precipitation nowcasting metrics. The results of the experiments conducted on these two datasets lead to the following findings:

- ConvLSTM is better than FC-LSTM in handling spatiotemporal correlations.
- Making the size of state-to-state convolutional kernel bigger than 1 is essential for capturing the spatiotemporal motion patterns.
- Deeper models can produce better results with fewer parameters.
- ConvLSTM performs better than ROVER for precipitation nowcasting.

Our implementations of the models are in Python with the help of Theano [3, 1]. We run all the experiments on a computer with a single NVIDIA K20 GPU. Also, more illustrative "gif" examples are included in the appendix.

### 4.1  Moving-MNIST Dataset

For this synthetic dataset, we use a generation process similar to that described in [21]. All data instances in the dataset are 20 frames long (10 frames for the input and 10 frames for the prediction) and contain two handwritten digits bouncing inside a $64 \times 64$ patch. The moving digits are chosen randomly from a subset of 500 digits in the MNIST dataset.[3] The starting position and velocity direction are chosen uniformly at random and the velocity amplitude is chosen randomly in $[3, 5)$. This generation process is repeated 15000 times, resulting in a dataset with 10000 training sequences, 2000 validation sequences, and 3000 testing sequences. We train all the LSTM models by minimizing the cross-entropy loss[4] using back-propagation through time (BPTT) [2] and

Table 1: **Comparison of ConvLSTM networks with FC-LSTM network on the Moving-MNIST dataset.** '-5x5' and '-1x1' represent the corresponding state-to-state kernel size, which is either $5 \times 5$ or $1 \times 1$. '256', '128', and '64' refer to the number of hidden states in the ConvLSTM layers. '(5x5)' and '(9x9)' represent the input-to-state kernel size.

| Model | Number of parameters | Cross entropy |
|---|---|---|
| FC-LSTM-2048-2048 | 142,667,776 | 4832.49 |
| ConvLSTM(5x5)-5x5-256 | 13,524,496 | 3887.94 |
| ConvLSTM(5x5)-5x5-128-5x5-128 | 10,042,896 | 3733.56 |
| **ConvLSTM(5x5)-5x5-128-5x5-64-5x5-64** | **7,585,296** | **3670.85** |
| ConvLSTM(9x9)-1x1-128-1x1-128 | 11,550,224 | 4782.84 |
| ConvLSTM(9x9)-1x1-128-1x1-64-1x1-64 | 8,830,480 | 4231.50 |

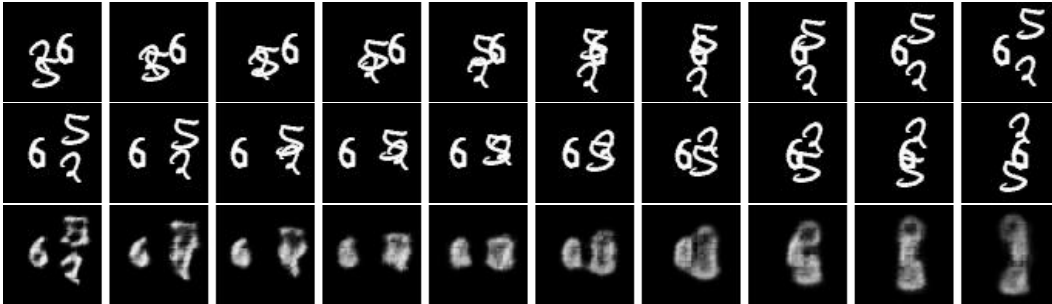

Figure 4: **An example showing an "out-of-domain" run.** From left to right: input frames; ground truth; prediction by the 3-layer network.

*RMSProp* [24] with a learning rate of $10^{-3}$ and a decay rate of 0.9. Also, we perform early-stopping on the validation set.

Despite the simple generation process, there exist strong nonlinearities in the resulting dataset because the moving digits can exhibit complicated appearance and will occlude and bounce during their movement. It is hard for a model to give accurate predictions on the test set without learning the inner dynamics of the system.

For the FC-LSTM network, we use the same structure as the unconditional future predictor model in [21] with two 2048-node LSTM layers. For our ConvLSTM network, we set the patch size to $4 \times 4$ so that each $64 \times 64$ frame is represented by a $16 \times 16 \times 16$ tensor. We test three variants of our model with different number of layers. The 1-layer network contains one ConvLSTM layer with 256 hidden states, the 2-layer network has two ConvLSTM layers with 128 hidden states each, and the 3-layer network has 128, 64, and 64 hidden states respectively in the three ConvLSTM layers. All the input-to-state and state-to-state kernels are of size $5 \times 5$. Our experiments show that the ConvLSTM networks perform consistently better than the FC-LSTM network. Also, deeper models can give better results although the improvement is not so significant between the 2-layer and 3-layer networks. Moreover, we also try other network configurations with the state-to-state and input-to-state kernels of the 2-layer and 3-layer networks changed to $1 \times 1$ and $9 \times 9$, respectively. Although the number of parameters of the new 2-layer network is close to the original one, the result becomes much worse because it is hard to capture the spatiotemporal motion patterns with only $1 \times 1$ state-to-state transition. Meanwhile, the new 3-layer network performs better than the new 2-layer network since the higher layer can see a wider scope of the input. Nevertheless, its performance is inferior to networks with larger state-to-state kernel size. This provides evidence that larger state-to-state kernels are more suitable for capturing spatiotemporal correlations. In fact, for $1 \times 1$ kernel, the receptive field of the states will not grow as time advances. But for larger kernels, later states have larger receptive fields and are related to a wider range of the input. The average cross-entropy loss (cross-entropy loss per sequence) of each algorithm on the test set is shown in Table 1. We need to point out that our experiment setting is different from [21] where an infinite number of training data is assumed to be available. The current offline setting is chosen in order to understand how different models perform in occasions where not so much data is available. Comparison of the 3-layer ConvLSTM and FC-LSTM in the online setting is included in the appendix.

Next, we test our model on some "out-of-domain" inputs. We generate another 3000 sequences of three moving digits, with the digits drawn randomly from a different subset of 500 MNIST digits that does not overlap with the training set. Since the model has never seen any system with three digits, such an "out-of-domain" run is a good test of the generalization ability of the model [21]. The average cross-entropy error of the 3-layer model on this dataset is 6379.42. By observing some of the prediction results, we find that the model can separate the overlapping digits successfully and predict the overall motion although the predicted digits are quite blurred. One "out-of-domain" prediction example is shown in Fig. 4.

## 4.2  Radar Echo Dataset

The radar echo dataset used in this paper is a subset of the three-year weather radar intensities collected in Hong Kong from 2011 to 2013. Since not every day is rainy and our nowcasting target is precipitation, we select the top 97 rainy days to form our dataset. For preprocessing, we first transform the intensity values $Z$ to gray-level pixels $P$ by setting $P = \frac{Z - \min\{Z\}}{\max\{Z\} - \min\{Z\}}$ and crop the radar maps in the central $330 \times 330$ region. After that, we apply the disk filter[5] with radius 10 and resize the radar maps to $100 \times 100$. To reduce the noise caused by measuring instruments, we further remove the pixel values of some noisy regions which are determined by applying $K$-means clustering to the monthly pixel average. The weather radar data is recorded every 6 minutes, so there are 240 frames per day. To get disjoint subsets for training, testing and validation, we partition each daily sequence into 40 non-overlapping frame blocks and randomly assign 4 blocks for training, 1 block for testing and 1 block for validation. The data instances are sliced from these blocks using a 20-frame-wide sliding window. Thus our radar echo dataset contains 8148 training sequences, 2037 testing sequences and 2037 validation sequences and all the sequences are 20 frames long (5 for the input and 15 for the prediction). Although the training and testing instances sliced from the same day may have some dependencies, this splitting strategy is still reasonable because in real-life nowcasting, we do have access to all previous data, including data from the same day, which allows us to apply online fine-tuning of the model. Such data splitting may be viewed as an approximation of the real-life "fine-tuning-enabled" setting for this application.

We set the patch size to 2 and train a 2-layer ConvLSTM network with each layer containing 64 hidden states and $3 \times 3$ kernels. For the ROVER algorithm, we tune the parameters of the optical flow estimator[6] on the validation set and use the best parameters (shown in the appendix) to report the test results. Also, we try three different initialization schemes for ROVER: ROVER1 computes the optical flow of the last two observed frames and performs semi-Lagrangian advection afterwards; ROVER2 initializes the velocity by the mean of the last two flow fields; and ROVER3 gives the initialization by a weighted average (with weights 0.7, 0.2 and 0.1) of the last three flow fields. In addition, we train an FC-LSTM network with two 2000-node LSTM layers. Both the ConvLSTM network and the FC-LSTM network optimize the cross-entropy error of 15 predictions.

We evaluate these methods using several commonly used precipitation nowcasting metrics, namely, rainfall mean squared error (Rainfall-MSE), critical success index (CSI), false alarm rate (FAR), probability of detection (POD), and correlation. The Rainfall-MSE metric is defined as the average squared error between the predicted rainfall and the ground truth. Since our predictions are done at the pixel level, we project them back to radar echo intensities and calculate the rainfall at every cell of the grid using the Z-R relationship [15]: $Z = 10 \log a + 10b \log R$, where $Z$ is the radar echo intensity in dB, $R$ is the rainfall rate in mm/h, and $a, b$ are two constants with $a = 118.239, b = 1.5241$. The CSI, FAR and POD are skill scores similar to precision and recall commonly used by machine learning researchers. We convert the prediction and ground truth to a 0/1 matrix using a threshold of 0.5mm/h rainfall rate (indicating raining or not) and calculate the hits (prediction = 1, truth = 1), misses (prediction = 0, truth = 1) and false alarms (prediction = 1, truth = 0). The three skill scores are defined as $\text{CSI} = \frac{\text{hits}}{\text{hits} + \text{misses} + \text{falsealarms}}$, $\text{FAR} = \frac{\text{falsealarms}}{\text{hits} + \text{falsealarms}}$, $\text{POD} = \frac{\text{hits}}{\text{hits} + \text{misses}}$. The correlation of a predicted frame $P$ and a ground-truth frame $T$ is defined as $\frac{\sum_{i,j} P_{i,j} T_{i,j}}{\sqrt{(\sum_{i,j} P_{i,j}^2)(\sum_{i,j} T_{i,j}^2)} + \varepsilon}$ where $\varepsilon = 10^{-9}$.

Table 2: Comparison of the average scores of different models over 15 prediction steps.

| Model | Rainfall-MSE | CSI | FAR | POD | Correlation |
|---|---|---|---|---|---|
| **ConvLSTM(3x3)-3x3-64-3x3-64** | **1.420** | **0.577** | **0.195** | **0.660** | **0.908** |
| Rover1 | 1.712 | 0.516 | 0.308 | 0.636 | 0.843 |
| Rover2 | 1.684 | 0.522 | 0.301 | 0.642 | 0.850 |
| Rover3 | 1.685 | 0.522 | 0.301 | 0.642 | 0.849 |
| FC-LSTM-2000-2000 | 1.865 | 0.286 | 0.335 | 0.351 | 0.774 |

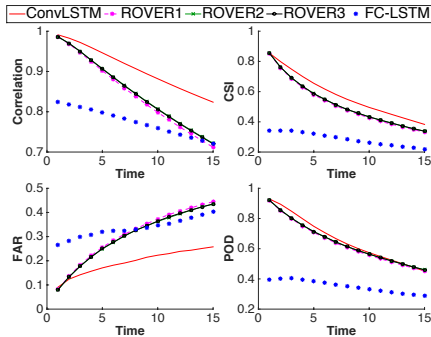

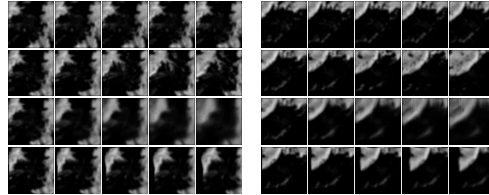

Figure 5: Comparison of different models based on four precipitation nowcasting metrics over time.

Figure 6: **Two prediction examples for the precipitation nowcasting problem.** All the predictions and ground truths are sampled with an interval of 3. From top to bottom: input frames; ground truth frames; prediction by ConvLSTM network; prediction by ROVER2.

All results are shown in Table 2 and Fig. 5. We can find that the performance of the FC-LSTM network is not so good for this task, which is mainly caused by the strong spatial correlation in the radar maps, i.e., the motion of clouds is highly consistent in a local region. The fully-connected structure has too many redundant connections and makes the optimization very unlikely to capture these local consistencies. Also, it can be seen that ConvLSTM outperforms the optical flow based ROVER algorithm, which is mainly due to two reasons. First, ConvLSTM is able to handle the boundary conditions well. In real-life nowcasting, there are many cases when a sudden agglomeration of clouds appears at the boundary, which indicates that some clouds are coming from the outside. If the ConvLSTM network has seen similar patterns during training, it can discover this type of sudden changes in the encoding network and give reasonable predictions in the forecasting network. This, however, can hardly be achieved by optical flow and semi-Lagrangian advection based methods. Another reason is that, ConvLSTM is trained end-to-end for this task and some complex spatiotemporal patterns in the dataset can be learned by the nonlinear and convolutional structure of the network. For the optical flow based approach, it is hard to find a reasonable way to update the future flow fields and train everything end-to-end. Some prediction results of ROVER2 and ConvLSTM are shown in Fig. 6. We can find that ConvLSTM can predict the future rainfall contour more accurately especially in the boundary. Although ROVER2 can give sharper predictions than ConvLSTM, it triggers more false alarms and is less precise than ConvLSTM in general. Also, the blurring effect of ConvLSTM may be caused by the inherent uncertainties of the task, i.e, it is almost impossible to give sharp and accurate predictions of the whole radar maps in longer-term predictions. We can only blur the predictions to alleviate the error caused by this type of uncertainty.

## 5 Conclusion and Future Work

In this paper, we have successfully applied the machine learning approach, especially deep learning, to the challenging precipitation nowcasting problem which so far has not benefited from sophisticated machine learning techniques. We formulate precipitation nowcasting as a spatiotemporal sequence forecasting problem and propose a new extension of LSTM called ConvLSTM to tackle the problem. The ConvLSTM layer not only preserves the advantages of FC-LSTM but is also suitable for spatiotemporal data due to its inherent convolutional structure. By incorporating ConvLSTM into the encoding-forecasting structure, we build an end-to-end trainable model for precipitation nowcasting. For future work, we will investigate how to apply ConvLSTM to video-based action recognition. One idea is to add ConvLSTM on top of the spatial feature maps generated by a convolutional neural network and use the hidden states of ConvLSTM for the final classification.

## Footnotes

[1]In real-life systems, radar echo maps are often constant altitude plan position indicator (CAPPI) images [9].

[2]It is worth noting that our precipitation nowcasting problem is different from the one studied in [14], which aims at predicting only the central region of just the next frame.

[3]MNIST dataset: http://yann.lecun.com/exdb/mnist/

[4]The cross-entropy loss of the predicted frame $P$ and the ground-truth frame $T$ is defined as $-\sum_{i,j,k} T_{i,j,k} \log P_{i,j,k} + (1 - T_{i,j,k}) \log(1 - P_{i,j,k})$.

[5]The disk filter is applied using the MATLAB function `fspecial('disk', 10)`.

[6]We use an open-source project to calculate the optical flow: `http://sourceforge.net/projects/varflow/`

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
