[Supplementary Material]

# Appendix of "Convolutional LSTM Network: A Machine Learning Approach for Precipitation Nowcasting"

Table 1: Best parameters for the optical flow estimator in ROVER.

| Parameter | Meaning | Value |
|:---:|:---:|:---:|
| $L_{max}$ | Coarsest spatial scale level | 6 |
| $L_{start}$ | Finest spatial scale level | 0 |
| $n_{pre}$ | Number of pre-smoothing steps | 2 |
| $n_{post}$ | Number of post-smoothing steps | 2 |
| $\rho$ | Gaussian convolution parameter for local vector field smoothing | 1.5 |
| $\alpha$ | Regularization parameter in the energy function | 2000 |
| $\sigma$ | Gaussian convolution parameter for image smoothing | 4.5 |

Figure 1: (**Larger Version**) Comparison of different models based on four precipitation nowcasting metrics over time.

Figure 2: (**Larger Version**) **Two prediction examples for the precipitation nowcasting problem.** All the predictions and ground truths are sampled with an interval of 3. From top to bottom: input frames; ground truth; prediction by ConvLSTM network; prediction by ROVER2.

Figure 3: **An illustrative example showing the in-domain prediction results of different models.** From top to bottom: input frames; ground truth; FC-LSTM; ConvLSTM-5X5-5X5-1-layer; ConvLSTM-5X5-5X5-2-layer; ConvLSTM-5X5-5X5-3-layer; ConvLSTM-9X9-1X1-2-layer; ConvLSTM-9X9-1X1-3-layer.

Figure 4: (**Larger Version**) **An illustrative example showing an out-domain run.** From top to bottom: input frames; ground truth; predictions of the 3-layer network.

Figure 5: **Comparison of the 3-layer ConvLSTM and FC-LSTM in the online setting.** In each iteration, we generate a new set of training samples and record the average cross entropy of that mini-batch. The $x$-axis is the number of data cases (starting from 25600) and the $y$-axis is the average cross entropy of the mini-batches. We can find that the loss of ConvLSTM decreases faster than FC-LSTM.