[Reviews · NeurIPS 2015]

Submitted by Assigned_Reviewer_1

The grid LSTM would be worth looking at as well: (http://arxiv.org/abs/1507.01526)
Summary: This is a well composed and neat application paper, applying a novel variant of the Long Short Term Memory network to a precipitation forecasting problem.

The authors do an excellent job of motivating the problem and use the LSTM to outperform the state-of-the-art approach from the application field.

Their version of the LSTM incorporates convolutions instead of standard fully connected layers, which seems novel and sensible.

Submitted by Assigned_Reviewer_2

A convolutional LSTM feels like one of those obvious-in-hindsight ideas, but to the best of my knowledge no one has published one yet and I commend the authors for finding an interesting and appropriate problem to apply them to.

I am concerned about the bouncing MNIST comparison to [21].

At first glance it appears the results in this paper are significantly worse than in [21]; [21] reports cross entropy of the unconditional future predictor model as 3520 (I'm assuming they average over the 10 predicted frames), which is quite a bit lower than the best result in this paper.

There is an explanation on line 321 which indicates that this is because [21] does not split into train and test sets, but I'm not convinced this is a proper criticism.

The data set described in [21] is generated online, and it is very unlikely that any of the train and test sequences are identical.

Due to the (clearly intentional) similarity of the bouncing MNIST experiment to the one conducted in [21] I think it would be more sensible to follow their setup more closely. As it stands it is not clear how to compare these two experiments because of the differences in how data is generated; I can't tell if the higher cross entropy in this paper is an artifact of the restricted training set or is a result of poor execution.

Criticisms of MNIST aside, I would have preferred to see more focus on the nowcasting problem in the experiments section.

Spending less time on MNIST videos and more time explaining Figure 6 (which I do not understand at all) would make the paper much stronger. More illustrations of phenomena that the ConvLSTM captures where other nowcasting methods fail (e.g. line 407) would be welcome.

It would be interesting to consider also reconstructing the reverse input sequence jointly with future prediction as is done in [21].

Line 76 states that [18] does not consider convolutional state to state transitions, however the rCNN model from that paper does use 1x1 recurrent convolutions.
Summary: This paper introduces the convolutional LSTM and applies it to an interesting problem.

Submitted by Assigned_Reviewer_3

This paper extends the recently proposed seq2seq encoder/decoder approach: the sequence of words is replaced by a sequence of images, and convolutions are used between several layers of a stacked LSTM and between the internal layers. This is a quite interesting application for the seq2seq framework, but I wonder whether this can be used for longer sequences, i.e. 20-100 images. The consecutive application of a convolution every time step could potentially lead to degenerated images (e.g. with a convolution that applies an average operator). On the other hand, I can imagine operators to detect movements of group of pixels which seems to be needed for the selected applications. The experimental results seem to indicate that the approaches works well in practice.
Summary: Prediction of a sequence of images given a sequence of images using convolution LSTM.

Submitted by Assigned_Reviewer_4

The paper describes an extension to the fully connected long short-term memory models for the purpose of prediction of doppler radar echo for a region over a short period of time given the recent doppler radar echo images.

The proposed framework extends LSTMs to handle the temporal evolution of spatial fields.

For the experiments, the paper considers two set ups, an artificial one based on the movements of handwritten digits, and a data set of Doppler radar images for unspecified region and time.

Quality: The proposed extension ConvLSTM appears to be mathematically sound and appropriate for the modeling of regularly timed spatio-temporal data.

The paper provides only a basic formulation of the framework; there are few details on the training of the model -- it is assumed that the reader is familiar with how FC-LSTMs are trained.

How does the training of ConvLSTM differ?

There are too few details provided about the precipitation data to judge whether the set up is appropriate.

It appears to be somewhat artificial to me in the following aspects: (1) storms can be fast moving (e.g., Midwest of the United States), so including the data only for the subregion of prediction may ignore the crucial information about the storm if it falls outside of the prediction area; (2) only days with precipitation are being used as a part of the data -- this appears unreasonable for a real application, (3) other covariates (e.g., time of day) are not taken into an account.

Additionally, to demonstrate an impact for the application, it would be appropriate to compare the predictive ability not just to other machine learning methods, but the standard methods in the application area, numerical weather prediction models (https://www.ncdc.noaa.gov/data-access/model-data/model-datasets/numerical-weather-prediction).

Clarity: The paper tries to cover quite a bit of ground, unfortunately, at the expense of clarity.

The details on the proposed model are fairly bare.

I would suggest to focus either on the application (and in this case, skip the dancing digits while elaborating more on the primary application) or on the methodology (and provide more details on the property of the new model at the expense of the details on the nowcasting application).

Originality: There are original contribution to the modeling framework and in applying it to the short-term precipitation forecasting.

Personally, I am glad to read that deep learning techniques are reaching precipitation modeling research area.

Significance: I do not have the expertise to judge whether the incremental leap between FC-LSTM and ConvLSTM is significant.

For the nowcasting application, while the proposed framework can be a useful intermediate step, due to the reservations mentioned under Quality section, I doubt it is close to solving the problem.

Minor: - Please check equation (3); I think some of the Hadamard products should actually be convolution operators. - Page 1, The acronym for numerical weather forecasting is NWF.

Did you mean numerical weather prediction? - Footnote on page 1, "plan"->"plane"?
Summary: The paper proposes an extension to a long short-term memory model to handle spatio-temporal data and attempts to apply this new framework for a short-term prediction of rainfall.

However, the set up is not convincing from the stand-point of application, and the paper is light on detail on the conceptual contribution; in the opinion of this reviewer, the paper would benefit from focusing on either the methodology or the application.

Author Feedback
Author rebuttal: We thank all reviewers for their constructive comments and suggestions. We will revise our paper accordingly.

For R1:
Thanks for your positive review.

Q1:...MNIST...focus...
Sorry for the confusion caused by our improper criticism of [21]. A limited number of training/testing split instead of online training is used in the paper to help understand the performance of different models in an offline setting where not so much data is available. We will emphasize this difference and remove some of the MNIST experiments to leave more space for the background and experiment analysis of the nowcasting problem in the revised paper.
In the online setting, after seeing 256000 sequences, the loss of ConvLSTM reaches 3400 while that of FC-LSTM is above 4000. It takes 1280000 cases for FC-LSTM to reach 3400.

Q2:Fig.6
Fig.6 shows two prediction examples generated by ConvLSTM and ROVER. Both models use the 5 observed radar echo maps to predict the future 15 echo maps. The top row shows the 5 input frames. The second row shows the ground truth frames sampled with an interval of 3 (resulting in 5 frames). The third and the last rows separately show the prediction results of ConvLSTM and ROVER (also sampled). In both examples, there is sudden agglomeration of rainfall in the top-left region. ConvLSTM has predicted some of these outside rainfalls while ROVER just blanks out them.

For R2:
Thanks for your review.

Q1:1D vector
We will revise our presentation to be more rigorous. Here we are trying to point out that FC-LSTM has not encoded any spatial information in its structure. For FC-LSTM, at a certain timestamp, every pair of input/state & state/state elements are connected and the state elements can be randomly permuted without affecting the overall topology, making the whole structure contain no spatial information. However, for spatiotemporal data studied in the paper, strong correlations exist between spatially and temporally nearby elements, which makes such structure not so appropriate.

Q2:Contribution
1) Our paper is the first to propose a variant of LSTM with convolutional recurrence. The difference between ConvLSTM and rCNN is that rCNN is based on vanilla RNN instead of LSTM and it constrains the state-to-state kernel size to 1X1 while we have shown that ConvLSTM can perform better with larger kernels (Section 4.1). For convolutional recurrence, there is a huge difference between 1X1 kernel and larger kernels. After unfolding along the temporal direction, convolutional recurrence becomes a deep convolutional neural network with shared kernels. For 1X1 kernel, the receptive field of states will not grow as time advances. While for larger kernels, later states (lying in deeper layers) will have wider receptive fields.
2) Our paper has built the first end-to-end trainable model for precipitation nowcasting. The model is different from [21] in that our building block is ConvLSTM and our task is spatiotemporal sequence forecasting instead of representation learning. Also, it is different from [18] in that [18] only considers one-step prediction while our target is multi-step and our model is end-to-end trainable.

For R3:
Thanks for your detailed review.

Q1:Training
We use BPTT + RMSProp (Line 262-264).

Q2:...subregion...
Our target application is precipitation nowcasting for the future 1.5 hours over a local city (Section 2.1 & 4.2). For this problem, the operational system SWIRLS which we compare uses only the past radar echo maps of the same region to nowcast the future. We agree that including information from the outside area will be useful, but it is currently difficult for us to obtain them. Also, since there are not so many rainy days in our data and our modeling target is precipitation, we select the rainy days.

Q3:Other covariates
This is a good future direction. In fact, the operational ROVER algorithm also has not included this type of information. Since our model is end-to-end trainable, additional features can be added as new channels to the input and we can train the network and get predictions using the same algorithm.

Q4:Comparison
In the paper, we have compared our model with the ROVER algorithm used in HKO. For NWP models, due to time required to gather observations, data assimilation and attaining physical balance among numerical processes, the first 1-2 hours of model forecasts may not be available or suitable for direct comparison with nowcasting products. We will explore this idea in the future.

Q5:Focus
We will remove some of the MNIST experiments to leave more space for the background and experiment analysis of the nowcasting problem and our ConvLSTM model in the revised paper.

Q6:Minor
1) In the first two lines of (2), operators before c_{t-1} should be Hadamard product and (3) is correct
2) We mean numerical weather prediction
3) Should be plan

For R4, R5, R6:
Thanks for your positive reviews. We will experiment on longer sequences in the future.